# Curcumin Remedies Testicular Function and Spermatogenesis in Male Mice with Low-Carbohydrate-Diet-Induced Metabolic Dysfunction

**DOI:** 10.3390/ijms231710009

**Published:** 2022-09-02

**Authors:** Chih-Wei Tsao, Pei-Shan Ke, Hsin-Yi Yang, Ting-Chia Chang, Chin-Yu Liu

**Affiliations:** 1Division of Urology, Department of Surgery, Tri-Service General Hospital, National Defense Medical Center, Taipei 114202, Taiwan; 2Division of Experimental Surgery Center, Department of Surgery, Tri-Service General Hospital, National Defense Medical Center, Taipei 114202, Taiwan; 3Department of Nutritional Science, Fu Jen Catholic University, New Taipei City 242304, Taiwan

**Keywords:** low-carbohydrate diet, curcumin, oxidative stress, male infertility, spermatogenesis

## Abstract

Increasing reports on the significance of dietary patterns in reproduction have arisen from both animal and human studies, suggesting an interactive association between nutrition and male fertility. The aim of this study was to investigate the effects of curcumin supplementation on low-carbohydrate-diet-induced metabolic dysfunction, testicular antioxidant capacity, apoptosis, inflammation and spermatogenesis in male mice. Male C57BL/6 mice were fed a normal diet (AIN-93M group, n = 12) and a low-carbohydrate diet for 12 weeks (LC group, fed with low-carbohydrate diet, n = 48), and mice randomly chosen from the LC group were later fed their original diet (LC group, n = 12). This diet was changed to AIN-93M feed (LC/AIN-93M group, n = 12), a ketogenic diet (LC/KD group, n = 12), or a ketogenic diet treated with curcumin supplementation for the final 6 weeks (LC/KDCu group, n = 12). A poor sperm morphology and mean testicular biopsy score (MTBS) were observed in the LC and LC/KD groups, but they were eliminated by the normal diet or ketogenic diet with curcumin. The LC group exhibited a lower testicular testosterone level and a lower 17β-HSD activity and protein expression. This also enhanced apoptosis protein expressions in testis tissue, including Bax/BCl_2_, cleaved caspase 3, PARP and NF-κB. Meanwhile, we found a statistically significant increase in lipid peroxidation and decreased superoxide dismutase (SOD), catalase (CAT) and glutathione peroxidase levels in the LC group. Our study indicated that a replacement of a normal diet or ketogenic diet supplemented with curcumin attenuated poor semen quality and reduced testosterone levels by the LC diet by reducing oxidative stress.

## 1. Introduction

Infertility is a health problem worldwide, affecting 15–20% of couples, i.e., about 70 million couples [1,2]. A male factor, including decreased sperm quality, is solely responsible for infertility in 20–30% of infertile couples, but contributes to infertility in 50% of couples overall [3,4]. Several studies have indicated that diet could play important roles in modulating spermatogenesis, sperm maturation and fertilizing ability [5].

Evidence regarding the relationships between health outcomes and long-term dietary fat intake has caused some controversy. In the past, the trend was to limit fat intake, especially saturated fatty acids [6]; however, current dietary guidelines emphasize that the focus should be on dietary patterns rather than particular levels of nutrients [7]. Dietary patterns are defined as non-random combinations, proportions or types of different foods and beverages [8]. Increasingly popular dietary patterns include the low-carbohydrate diet, low-fat diet and the Mediterranean diet [7]. The definition of a low-carbohydrate diet is <26% or <130 gm/day carbohydrate intake and limited refined carbohydrates [9,10].

A cross-sectional study analyzed the relationships between dietary patterns and sperm quality, and the results show that men who consumed a Western diet tend to have a poor sperm concentration and an abnormal sperm morphology. Men who consumed a high-carbohydrate diet exhibited significant decreases in total sperm motility and progressive motility, and an abnormal sperm morphology was observed in men who consumed a high-sodium diet. Moreover, higher intakes of high-sugar snacks and sugar-sweetened drinks tend to cause a much lower sperm concentration [11]. Eslamian et al. reported that high consumptions of red meat and sugar-sweetened beverages are associated with a greater risk of asthenozoospermia, and conversely, high fruit, vegetable, poultry, skimmed milk and seafood intake are associated with a lower risk of sperm abnormality [2,12].

The most common cause of male infertility is defective sperm function, and oxidative stress due to increased lipid peroxidation or a lack of antioxidants in sperm is one of the major contributing factors [13]. Sperm motility and hyperactivation require substantial ATP produced by glycolysis and oxidative phosphorylation that mainly uses glycolysable sugar such as glucose, mannose, and fructose [14,15]. Tanaka et al. found that testis-specific succinyl CoA transferase (SCOT-t) may be functional in sperm for the utilization of ketone bodies [16].

Curcumin (1,7-bis (4-hydroxy-3-methoxyphenyl)-1,6-hepadiene-3,5-dione), a powdered rhizome of *Curcuma longa*, is considered a powerful antioxidant. The antioxidant capacity of curcumin can be divided into the direct removal of reactive oxygen species (ROS) and indirect activation of antioxidant enzymes activities [17,18]. The former involves the direct scavenging of superoxide anions (O_2_^·^) [19], hydroxyl radicals (∙OH) [20], nitric oxide (NO), and peroxyl radicals (ROO∙) [21], while the latter involves the induction of superoxide dismutase (SOD), catalase (CAT) [22], glutathione reductase (GR) [20], glutathione peroxidase (GPx) and glutathione S-transferase (GST) activity [23]. These properties are related to the chemical structure of curcumin, which includes bios-α, β-unsaturated β-diketone, two methoxy groups, two phenolic hydroxy groups and two conjugated bonds, which might play important roles in anti-inflammatory and antiproliferative activities [24,25]. A randomized, double-blind, placebo-controlled clinical trial showed that curcumin supplementation could increase sperm quality, including total sperm count, sperm concentration and motility, and improved the total antioxidant capacity of plasma, malondialdehyde, C-reactive protein and tumor necrosis factor (TNF) [26].

The low-carbohydrate diet is a fashionable strategy for weight control and exercise performance, and we carried out this study to explore whether this diet can reduce the impact of obesity and improve male reproductive function and, if possible, to identify the related mechanisms. The present study aimed to investigate the effects of a ketogenic diet and curcumin supplementation on reproductive damage induced by a low-carbohydrate diet and its involved mechanisms, including oxidative stress, apoptosis and inflammation.

## 2. Results

### 2.1. Curcumin Does Not Affect Reproductive Organ Weight and Lipid Profiles

As shown in Figure 1, daily food intake and body weight were not statistically different among the five groups, which may be ascribed to pair-feeding. Mice that were fed a low-carbohydrate diet (LC) had significantly higher levels of total cholesterol, TG and HDL. The LC that switched to the AIN-93M diet group (LC/AIN-93M) had higher AST and ALT levels compared with the LC group. In addition, improved serum TG and HDL levels were observed in the LC that switched to the ketogenic diet in the curcumin supplementation group (LC/KDCu).

### 2.2. Curcumin Ameliorates the Decrease in Testicular Spermatogenesis and Sperm Quality Caused by a Low-Carbohydrate Diet

The results show that there were no significant differences in the weights of the testis, epididymis or vas deferens among the five groups. In terms of sperm motility, the AIN-93M group was significantly better than the other four groups, and regarding sperm count, there were no significant differences among the five groups. In the case of sperm morphology, the percentage of sperm of a normal morphology in the LC group was significantly lower than in the other four groups (Figure 2). Figure 3 illustrates the correlation between ketone bodies and sperm quality. Correlation results are as follows: sperm motility (*p* = 0.0043, r = 0.1490), sperm count (*p* = 0.2037, r = 0.0292), and sperm morphology (*p* = 0.7551, r = 0.0055). Figure 4 shows that there was no significant difference in the MSTD among the five groups. The MTBS values of the LC and the LC that switched to the ketogenic diet (LC/KD) groups were significantly lower than those of the AIN-93M and LC/KDCu groups, and the thickness of the GE in the LC and LC/KD groups was significantly lower than in the other three groups. To sum up, the LC/AIN-93M and the LC/KDCu groups had better sperm production and improved spermatogenesis.

### 2.3. Curcumin Increases the Testosterone Level through Upregulating 17β-HSD Expression

The testicular testosterone, 3β-HSD and 17β-HSD levels were measured by ELISA. The results show that the LC and LC/KD groups had significantly lower testosterone concentrations than the AIN-93M group. Additionally, the LC group had a lower 17β-HSD level than the other groups. In contrast, the LC/KDCu group had an increased testosterone concentration and 17β-HSD level (Figure 5). The protein expressions of enzymes involved in the testosterone biosynthesis pathway are shown in Figure 6, and there were no statistical differences in the expressions of StAR, CYP11A1, 3β-HSD or CYP17A1. The LC group had a significantly lower 17β-HSD expression than the AIN-93M group, whereas the LC/AIN-93M and LC/KDCu groups had significantly increased expressions.

### 2.4. Curcumin Activates Antioxidant Capacity and Suppresses Lipid Peroxidation, Apoptosis and Inflammation in the Testis 

As shown in Figure 7, the levels of SOD, GPx and CAT in the LC group were significantly lower than in the other groups. In terms of lipid peroxidation, the MDA concentration in the LC group was significantly higher than in the other four groups. A low-carbohydrate diet resulted in higher protein expressions of the ratio of the pro-apoptotic factor Bax to the anti-apoptotic factor Bcl-xl (Bax:Bcl-xl), cleaved caspase 8, caspase 3 and PARP, while these changes were attenuated following a change in the diets of the AIN-93M and LC/KDCu groups (Figure 8 and Figure 9). The examination of inflammation-related factors showed an increasing NF-κB protein expression in the LC and LC/KD groups and a notably lower NF-κB expression in the LC/KDCu group (Figure 10).

## 3. Discussion

The relationship between diet and infertility is a major clinical and public health problem. In order to investigate the effects of different dietary patterns on male fertility, the present study used pair-feeding to ensure that food intake in the control group was matched to that in the experimental groups, so that the impacts of treatments were determined independently of food intake [27]. Li et al. performed a pair-feeding experiment, and no statistical differences in the initial body weight, final body weight or body weight gain among groups fed a liquid diet, high-carbohydrate liquid diet and high-fat liquid diet were found [28].

The results (Figure 1) show that low-carbohydrate diet administration significantly increased the serum levels of TC, TG and HDL, and decreased the AST and ALT levels, in comparison with the AIN-93M group. Volek et al. explained the lipoprotein metabolism when consuming a low-carbohydrate diet, which initially increases circulating TAG-rich chylomicrons, which are then cleared by lipoprotein lipase (LPL). A low-carbohydrate diet also decreases glucose and insulin levels, and then decreases LPL, and increases TG hydrolysis and fatty acid production. Then, circulating fatty acids are diverted away to TG and towards mitochondrial oxidation to acetyl CoA. Excessive mitochondrial oxidation eventually boosts the formation of ketone bodies. Unesterified cholesterol, phospholipid, and apolipoprotein are transferred from VLDL and form mature HDL-C [29].

In this study, we demonstrated that curcumin could ameliorate poor spermatogenesis and sperm function, and reverse oxidative stress, inhibit inflammation and inhibit apoptosis in the testes of low-carbohydrate-diet-fed mice. A prior study explored the mechanisms of male infertility and focused on sperm parameters or reproductive hormones. Sperm mainly use sugars as an energy source, including glucose, mannose, and fructose [14]. Attaman et al. found that a high intake of total fat was negatively related to sperm count and concentration [30]. Curcumin, a phenolic compound extracted from the *Curcuma longa* rhizome, has antioxidant, anti-inflammatory, and anti-mutative properties [31]. The present study shows that curcumin improved the percentage of morphologically normal sperm (Figure 2) and testicular morphology, and increased the thickness of the GE and the MTBS (Figure 4). Alizadeh et al. showed that curcumin could improve sperm count, concentration and motility in patients with asthenoteratospermia. Significant improvements in plasma levels of total antioxidant capacity, malondialdehyde and TNF were also observed [26]. It is speculated that curcumin, with a conjugated structure and an enol form, could scavenge free radicals and increase the activity of antioxidant enzymes, thereby improving sperm quality [19,32]. The study shows that oral curcumin (80 mg/kg) can lower lipid accumulation in liver and adipose tissue and improve the insulin sensitivity of male C57BL/6 mice with a 60% high-fat diet by regulating the SREBP pathway [33]. Prophylactic oral administration of curcumin (80 mg/kg) in Sprague Dawley rats with a 60% high-fat diet feeding showed anti-hyperglycemic, anti-lipolytic and anti-inflammatory effects by attenuating TNF-α levels [34]. In ICR mice with spermatogenic disorders induced by scrotal heat stress, administrating curcumin (80 mg/kg) by intragastric intubation, also had antioxidative, anti-apoptotic and androgen synthesis effects [35]. In our study, the low-carbohydrate diet and ketogenic diet contained 69.9% and 83.2% fat, respectively. To mimic the fashionable dietary patterns for weight control, we aimed to illustrate the effects of different dietary patterns and fat contents with or without curcumin on testicular antioxidant capacity, apoptosis, inflammation and spermatogenesis. Our results show that a low-carbohydrate diet caused a lower sperm quality and a damaged testicular histology (Figure 2 and Figure 4). Administrating curcumin (80 mg/kg) could partially reverse this condition.

The limiting factors of curcumin, such as a low solubility and oral bioavailability, restrict its application as a therapeutic agent [36,37] to develop oral delivery systems for curcumin to enhance solubility and oral bioavailability, such as nanoparticles [38]. The study shows that curcumin nanomicelle supplement (80 mg per day) to asthenoteratospermia [38] or oligozoospermia [26] patients could improve semen quality. In the present study, a low-carbohydrate diet impaired sperm motility, morphology, MTBS and GE, and the ketogenic diet with curcumin improved morphology, MTBS and GE but not including motility. Bioavailability of curcumin could be improved by nano-curcumin formulations. Therefore, future investigation is suggested in this regard.

The concentration of testosterone plays an important role in spermatogenesis [39]. In this study, we measured the level of testosterone and protein expressions of markers involved in the testosterone biosynthesis pathway. The data show that the testosterone concentration in the LC group was significantly lower than that in the AIN-93M group and increased in the LC/AIN-93M and LC/KDCu groups (Figure 5). Moreover, our results show that the LC group had a lower protein expression of 17β-hydroxysteroid dehydrogenase than the AIN-93M group, and this was increased in the LC/AIN-93M, LC/KD and LC/KDCu groups (Figure 5 and Figure 6). Sharma et al. established a rat model of exposure to insecticides/xenobiotics in which the animals received supplementation with curcumin and quercetin. It was demonstrated that curcumin and quercetin could protect against insecticides and xenobiotic-induced sperm count, testosterone concentration, activities of 3β-HSD and 17β-HSD, and oxidative damage [40], and the increases in lipid peroxidation and oxidative stress may have contributed to the decreases in sperm count and the concentration of testosterone. On the other hand, it is possible that hypothalamic–pituitary–gonadal axis negative feedback inhibition may partly be responsible for the reduced testosterone level.

The most common mechanisms of infertility include oxidative stress, inflammation, and apoptosis [41,42]. As shown in the present study, a low-carbohydrate diet decreased SOD, catalase and GPx activities, while the LC/AIN-93M, LC/KD and LC/KDCu groups exhibited increased antioxidant activities (Figure 7). Mu et al. showed that a high-fat diet increased oxidative stress and MDA content, and curcumin supplementation decreased the ROS level and lipid peroxidation production [43]. The Leydig cell is the main source of testosterone secretion and sperm production. Excessive oxidative stress is harmful to sperm survival, damaging the sperm cell membrane, breaking the double-stranded DNA of genetic material, and activating apoptotic mediators, cytochrome c, caspase 9 and caspase 3 [44]. In our study, the LC group exhibited increased protein expressions of the Bax:Bcl-xl ratio, cleaved caspase 8, cleaved caspase 3 and cleaved PARP, and these were improved in the LC/AIN-93M and LC/KDCu groups (Figure 8 and Figure 9). Mu et al. showed that curcumin reversed high-fat-diet-induced decreased expressions of Fas, Bax and cleaved caspase 3, and increased the expression of Bcl-xl [45]. Inflammation is a physiological defense mechanism that increases oxidative stress. Inflammation may induce monocytes to differentiate into macrophages and produce proinflammatory cytokines, including IL-1, IL-6, and TNF-α [46]. In our study, the LC and LC/KD groups exhibited increased NF-κB expression, while the expression was decreased in the LC/AIN-93M and LC/KDCu groups (Figure 10). Rosenbaum et al. also showed that a ketogenic diet could increase the C-reactive protein level [47].

## 4. Materials and Methods

### 4.1. Animals

All experimental procedures were approved by the Institutional Animal Care and Use Committee (IACUC; ethical code number: LAC-2010-0050) of Taipei Medical University (Taipei City, Taiwan). Male C57BL/6 mice (6-week-old, weighing 22–24 g) were obtained from the National Laboratory Animal Center (Taipei City, Taiwan) and were housed in standard cages (room temperature: 20–26°C; humidity: 55 ± 5%; 12 h light–dark cycle). The mice were pair-fed a liquid diet containing maltose and dextrin.

### 4.2. Study Design

In the present study, after 1 week of acclimatization, the male C57BL/6 mice were randomly divided into two groups as follows: the AIN-93M group (n = 12) and the LC group (n = 48). After 12 weeks, the AIN-93M group was maintained on the AIN-93M diet, and the LC group was randomly divided into four groups: one was maintained on the original diet (LC, n = 12), the second was changed to the AIN-93M diet (LC/AIN-93M, n = 12), the third changed to a ketogenic diet (LC/KD, n = 12), and the final group changed to a ketogenic diet with curcumin supplementation (LC/KDCu, n = 12) for 6 weeks. The curcumin powder was dissolved in the soybean oil and mixed in the liquid diet. The ingredients of each diet are shown in Table 1.

### 4.3. Histological Analysis

Formalin-fixed testis tissues were prepared and analyzed at the Department of Pathology of Cardinal Tien Hospital (New Taipei City, Taiwan), being embedded, cut and stained with hematoxylin and eosin (H and E) following standard protocols. The samples were observed and captured under a DM1000 microscope (Leica, Wetzlar, Germany). The thickness of the testicular germinal epithelium (GE) and the mean seminiferous tubule diameter (MSTD) were calculated across axes using Image J software (1.50, National Institutes of Health, Bethesda, MD, USA). The GE size was measured using the average length of a lumen and the length of a seminiferous tubule, and at least 10 sections were calculated in each mouse. Johnsen’s score was used to evaluate testicular spermatogenesis. The degree of testicular damages was tested using a 1–10-point scale ranging from the highest score of 10, standing for complete spermatogenesis, to 1, indicating no seminiferous epithelium.

### 4.4. Serum Analysis

After anaesthetization, blood from the heart sat for 20–30 min at room temperature before being centrifuged for 15 min at 3000 rpm and separated the resulting supernatant. Total cholesterol (TC, mg/dL), triglyceride (TG, mg/dL), high-density lipoprotein (HDL, mg/dL), aspartate aminotransferase (AST, U/L), and alanine aminotransferase (ALT, U/L) levels were measured using a chemistry analyzer (Modular P800, Roche, Basel, Switzerland). Serum ketone (mmol/L) and glucose (mg/dL) levels were measured using a GLUCOSURE POC system (ApexBio AB-302G, Hsinchu, Taipei, Taiwan). The serum insulin level (pmol/L) was determined by enzyme-linked immunosorbent assay (ELISA) using a commercial ELISA kit according to the manufacturer’s instructions (Mercodia Mouse Insulin ELISA, 10-1247-01, Uppsala, Sweden).

### 4.5. Sperm Quality Analysis

Sperm quality was assessed according to sperm motility, sperm count and morphological abnormality. Sperm motility was expressed as the percentage of motile sperm evaluated microscopically under a magnification of 40× in four random microscopic fields, and at least 200 sperm cells were counted. The sperm count was measured using an automated cell counter (TC20, Bio-Rad, Hercules, CA, USA).

For sperm morphology, sperm slides were fixed with methanol (Honeywell, Morris Plains, NJ, USA) and stained with an eosin Y (E4009, Sigma-Aldrich, Saint Louis, MO, USA) and ethanol (Bioman, Taipei City, Taiwan) mixture. Then, the slides were rinsed with 75% ethanol (Bioman) and dried. Finally, the percentage of morphologically normal sperm within at least 100 sperm cells was calculated.

### 4.6. Testicular Hormone, Antioxidants and Marker of Lipid Peroxidation Analysis 

Testis tissues were homogenized in ice-cold RIPA lysis buffer (Thermo Fisher Scientific, Waltham, MA, USA) containing protease inhibitors and phosphatase inhibitors and centrifuged at 14,000× *g* (4 °C) for 20 min. The testicular concentrations of testosterone (Cayman, Item No. 582701, Ann Arbor, MI, USA), 3β-HSD (MyBioSource, #3806055, San Diego, CA, USA) and 17β-HSD (MyBioSource, #3806056) were measured using an ELISA kit according to the manufacturer’s instructions. The antioxidants and marker of lipid peroxidation, including superoxide dismutase (SOD, U/mL; Cayman, Item No. 706002), catalase (CAT, µM; Cayman, Item No. 707002), glutathione peroxidase (GPx, nmol/min/mL; Cayman, Item No. 703102) and malondialdehyde (MDA, µM; Cayman, Item No. 10009055) were also evaluated using ELISA kits.

### 4.7. Western Blotting Analysis

Total protein was extracted from testis tissues using RIPA lysis buffer, and the concentrations determined using a detergent-compatible protein assay (Bio-Rad). Equal quantities of protein were resolved using sodium dodecyl sulfate (SDS)-polyacrylamide gels (Bioman) and then transferred onto polyvinylidene difluoride (PVDF) membranes (GE Healthcare, Freiburg, Germany). The PVDF membranes were blocked with 5% (*w*/*v*) dried milk with 0.1% Tween 20 in Tris-buffered saline (TBST) for 1 h and incubated overnight with the following primary antibodies: StAR (1:1000; sc-25806, Santa Cruz Biotechnology, Dallas, TX, USA), 3β-HSD (1:500; sc-28206, Santa Cruz Biotechnology), 17β-HSD (1:250; sc-135044, Santa Cruz Biotechnology), CYP11A1 (1:1000; sc-202456, Santa Cruz Biotechnology), CYP17A1 (1:1000; sc-66850, Santa Cruz Biotechnology), caspase 8 (1:1000; 59607, GeneTex, San Antonio, TX, USA), Bax (1:1000; 2772, Cell Signaling Technology, Danvers, MA, USA), caspase 9 (1:1000; 9508, Cell Signaling Technology), caspase 3 (1:500; 9662, Cell Signaling Technology), cleaved caspase 3 (1:250; 9664, Cell Signaling Technology), poly (ADP-ribose) polymerase (PARP;1:1000; 3542, Cell Signaling Technology), Bcl-xl (1:1000; ab32370, Abcam, Cambridge, MA, USA), peroxisome proliferator-activated receptor ϒ (PPARϒ; 1:1000; sc-7273, Santa Cruz Biotechnology), interleukin-6 (IL-6; 1:1000; sc-57315, Santa Cruz Biotechnology), TNF-α (1:1000; ab1793, Abcam) and nuclear factor-κB (NF-κB; 1:1000; E381, Abcam). After incubation, the membranes were washed with TBST and incubated with secondary antibodies anti-mouse IgG (1:5000; sc-2005, Santa Cruz Biotechnology) or anti-rabbit IgG (1:4000; sc-2054, Santa Cruz Biotechnology) for 1 h at room temperature. Finally, the membranes were visualized using an ECL kit (Omicsbio, Taipei City, Taiwan). The intensity of each band was quantified using Image J software and normalized to the expression of β-actin.

### 4.8. Statistical Analysis

All the data are presented as means ± SD. Data were analyzed using one-way analysis of variance (ANOVA) followed by Duncan’s new multiple range test. Differences were defined as statistically significant if the *p*-value was less than 0.05. Pearson’s test was used to analyze the correlation between variables. *p* < 0.05 was considered statistically significant. All statistical analyses were conducted using SAS software, version 9.4 (SAS Institute Inc., Cary, NC, USA).

## 5. Conclusions

In summary, the results of our study confirmed that a low-carbohydrate diet led to a lower sperm quality and damaged testicular histology. Supplementation with curcumin may improve the impaired sperm and testis function via decreasing oxidative stress, inflammation and apoptosis.

## Figures and Tables

**Figure 1 ijms-23-10009-f001:**
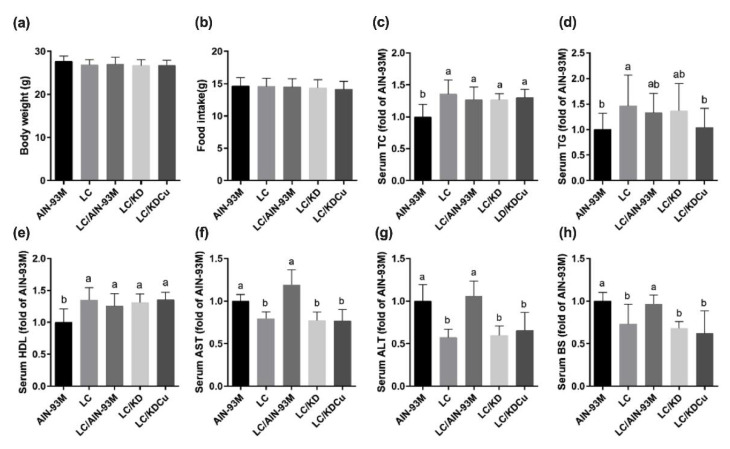
Effects of the low carbohydrate diet and curcumin on (**a**) body weight, (**b**) food intake, (**c**) serum total cholesterol (TC), (**d**) serum triglycerides (TG), (**e**) serum high density lipoprotein (HDL), (**f**) serum aspartate aminotransferase (AST), (**g**) serum alanine aminotransferase (ALT), (**h**) serum glucose in male mice. Data are expressed as means ± SD. ((**a**,**b**) n = 11–12; (**c**–**h**) n = 7–12 per group). Bars (a, b) with different letters indicate that values are significantly different. (One-way ANOVA, Duncan’s test, *p* < 0.05). AIN-93M: AIN-93M diet; LC: low carbohydrate diet; LC/AIN-93M: low carbohydrate diet changed to AIN-93M diet; LC/KD: low carbohydrate diet changed to ketogenic diet; LC/KDCu: low carbohydrate diet changed to ketogenic diet and curcumin.

**Figure 2 ijms-23-10009-f002:**
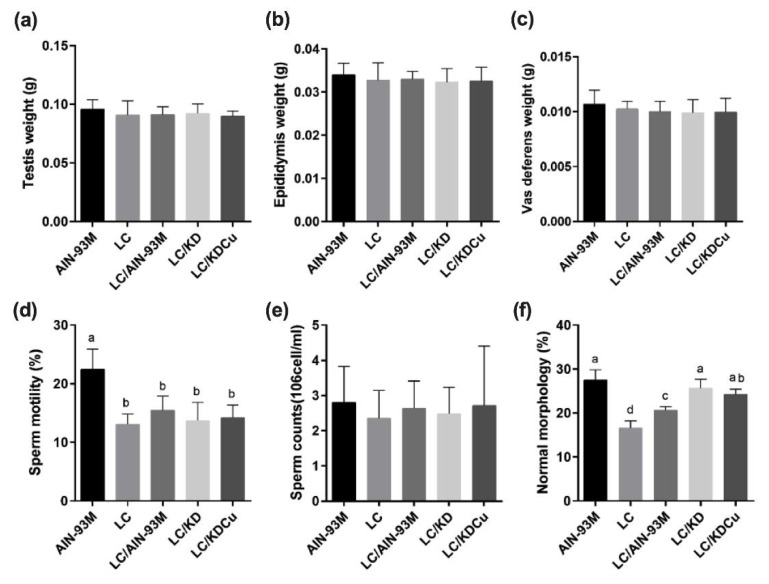
Effects of the low carbohydrate diet and curcumin on (**a**) testis, (**b**) epididymis, (**c**) vas deferens, (**d**) sperm motility, (**e**) sperm counts, (**f**) percentage of normal sperm morphology in male mice. Data are expressed as means ± SD. n = 11–12 per group. Bars (a, b, c, d) with different letters indicate that values are significantly different. (One-way ANOVA, Duncan’s test, *p* < 0.05). AIN-93M: AIN-93M diet; LC: low carbohydrate diet; LC/AIN-93M: low carbohydrate diet changed to AIN-93M diet; LC/KD: low carbohydrate diet changed to ketogenic diet; LC/KDCu: low carbohydrate diet changed to ketogenic diet and curcumin.

**Figure 3 ijms-23-10009-f003:**
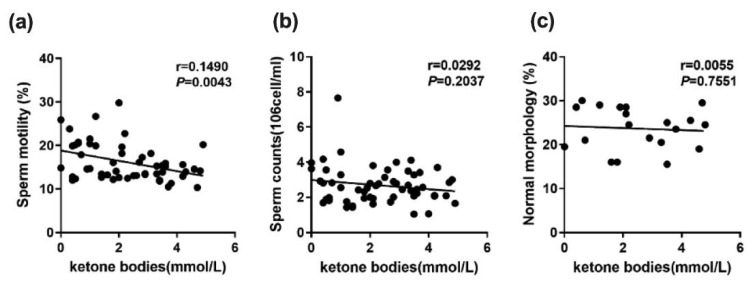
Correlation between ketone bodies and sperm quality parameters; (**a**) sperm motility, (**b**) sperm counts, (**c**) percentage of normal sperm morphology in male mice. n = 11–12 per group. (Pearson’s test, *p* < 0.05). r: spearman correlation coefficient.

**Figure 4 ijms-23-10009-f004:**
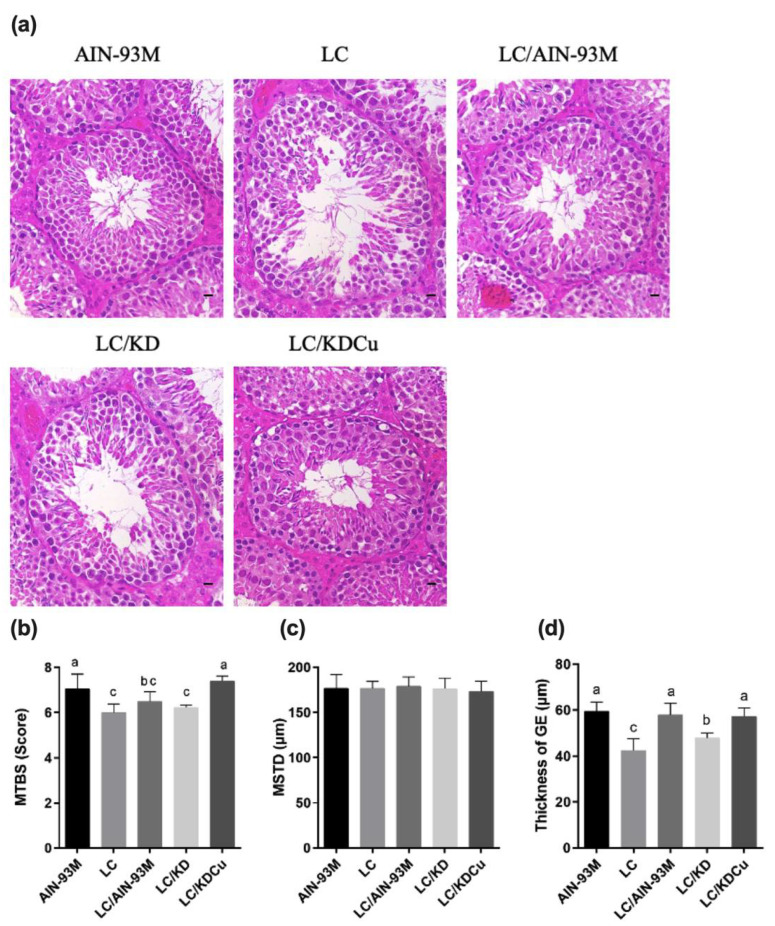
Effects of the low carbohydrate diet and curcumin on (**a**) testicular histology, (**b**) MTBS, (**c**) MSTD, (**d**) thickness of GE in male mice. Data are expressed as means ± SD. n = 5 per group. MTBS: mean testicular biopsy score; MSTD: mean seminiferous tubule diameter. Bars (a, b, c) with different letters indicate that values are significantly different. (One-way ANOVA, Duncan’s test, *p* < 0.05). AIN-93M: AIN-93M diet; LC: low carbohydrate diet; LC/AIN-93M: low carbohydrate diet changed to AIN-93M diet; LC/KD: low carbohydrate diet changed to ketogenic diet; LC/KDCu: low carbohydrate diet changed to ketogenic diet and curcumin.

**Figure 5 ijms-23-10009-f005:**
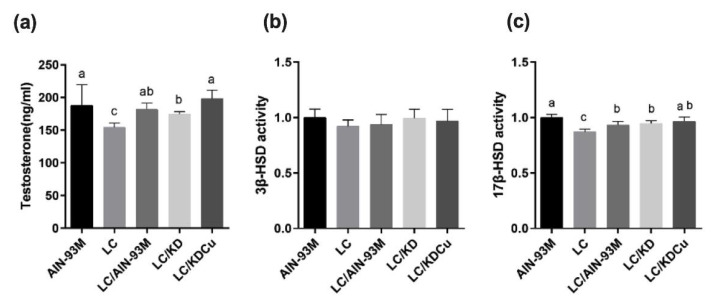
Effects of the low carbohydrate diet and curcumin on testicular (**a**) testosterone, (**b**) 3β-HSD, (**c**) 17β-HSD levels in male mice. Data are expressed as means ± SD. n = 5 per group. Bars (a, b, c) with different letters indicate that values are significantly different. (One-way ANOVA, Duncan’s test, *p* < 0.05). AIN-93M: AIN-93M diet; LC: low carbohydrate diet; LC/AIN-93M: low carbohydrate diet changed to AIN-93M diet; LC/KD: low carbohydrate diet changed to ketogenic diet; LC/KDCu: low carbohydrate diet changed to ketogenic diet and curcumin.

**Figure 6 ijms-23-10009-f006:**
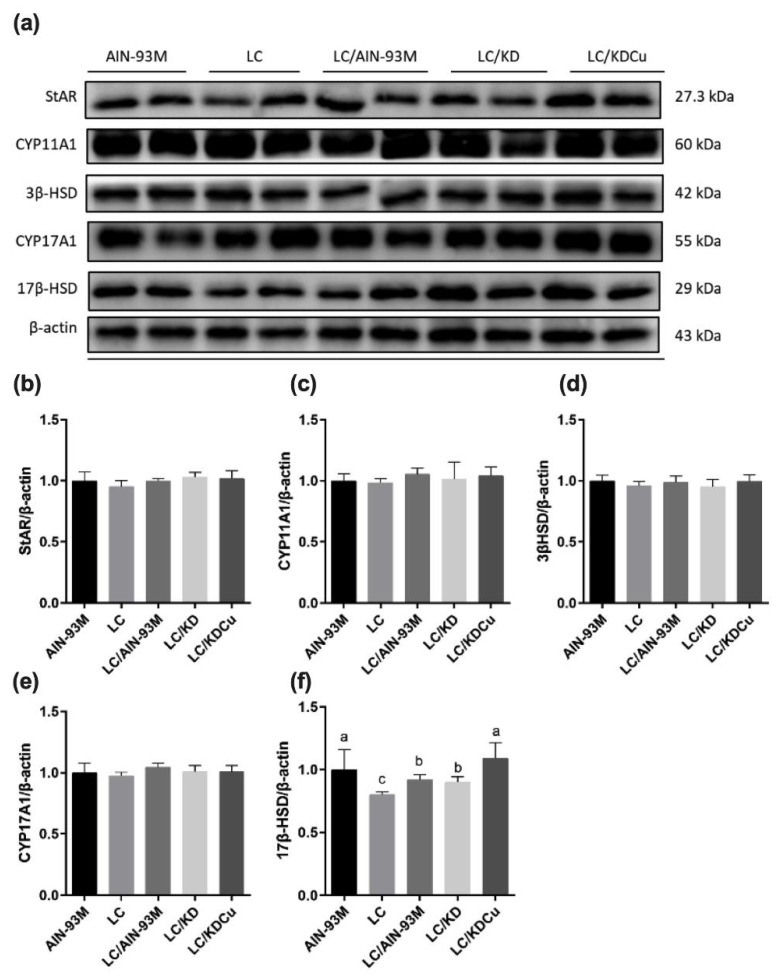
Effects of the low carbohydrate diet and curcumin on testosterone biosynthesis regulators (**a**) protein expressions in each group, (**b**) StAR, (**c**) CYP11A1, (**d**) 3β-HSD, (**e**) CYP17A1, (**f**) 17 β-HSD relative density analysis of the protein bands in male mice. Data are expressed as means ± SD. n = 6–9 per group. Bars (a, b, c) with different letters indicate that values are significantly different. (One-way ANOVA, Duncan’s test, *p* < 0.05). AIN-93M: AIN-93M diet; LC: low carbohydrate diet; LC/AIN-93M: low carbohydrate diet changed to AIN-93M diet; LC/KD: low carbohydrate diet changed to ketogenic diet; LC/KDCu: low carbohydrate diet changed to ketogenic diet and curcumin.

**Figure 7 ijms-23-10009-f007:**
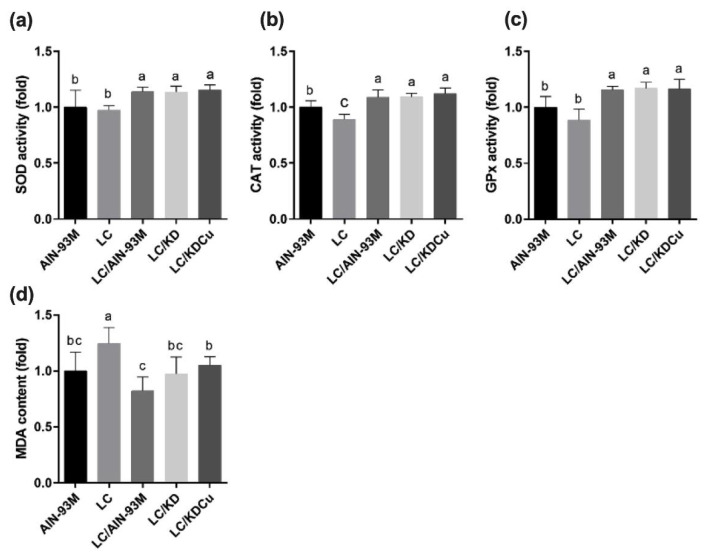
Effects of the low carbohydrate diet and curcumin on (**a**) SOD activity, (**b**) CAT activity, (**c**) GPx activity, (**d**) MDA content in male mice. Data are expressed as means ± SD. n = 6–9 per group. Bars (a, b, c) with different letters indicate that values are significantly different. (One-way ANOVA, Duncan’s test, *p* < 0.05). AIN-93M: AIN-93M diet; LC: low carbohydrate diet; LC/AIN-93M: low carbohydrate diet changed to AIN-93M diet; LC/KD: low carbohydrate diet changed to ketogenic diet; LC/KDCu: low carbohydrate diet changed to ketogenic diet and curcumin.

**Figure 8 ijms-23-10009-f008:**
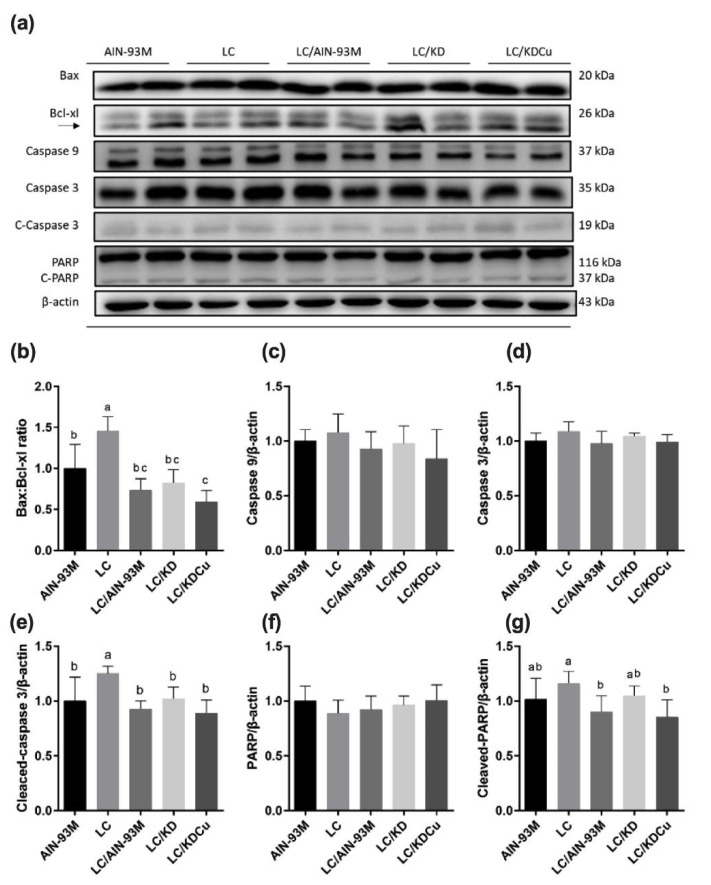
Effects of the low carbohydrate diet and curcumin on testicular intrinsic apoptosis pathway regulators (**a**) protein expressions in each group (**b**) Bax:Bcl-xl, (**c**) caspase 9, (**d**) caspase 3, (**e**) cleaved-caspase 3, (**f**) PARP, (**g**) cleaved-PARP in male mice. Data are expressed as means ± SD. n = 6–9 per group. Bars (a, b, c) with different letters indicate that values are significantly different. (One-way ANOVA, Duncan’s test, *p* < 0.05). AIN-93M: AIN-93M diet; LC: low carbohydrate diet; LC/AIN-93M: low carbohydrate diet changed to AIN-93M diet; LC/KD: low carbohydrate diet changed to ketogenic diet; LC/KDCu: low carbohydrate diet changed to ketogenic diet and curcumin.

**Figure 9 ijms-23-10009-f009:**
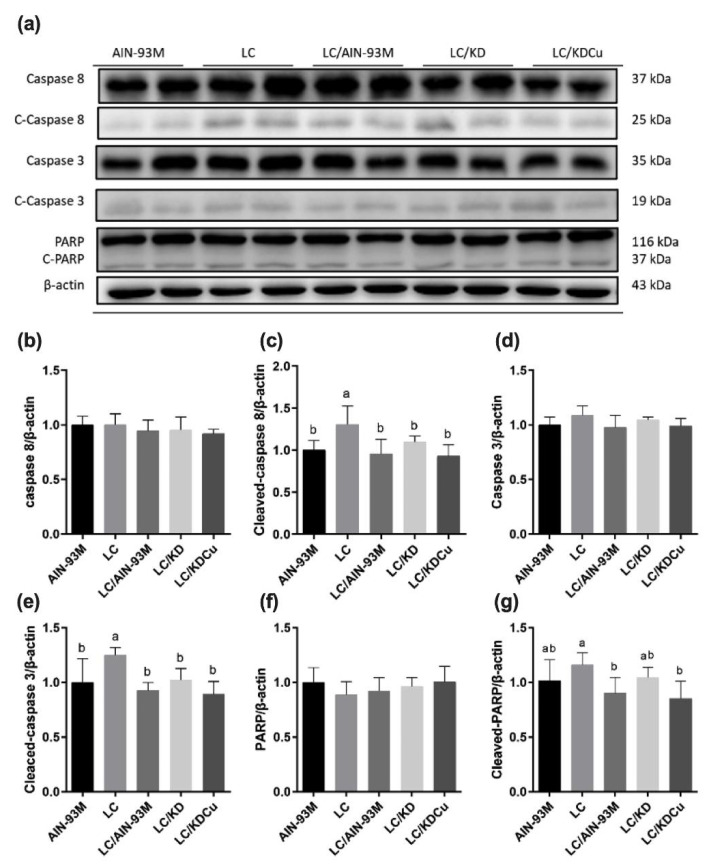
Effects of the low carbohydrate diet and curcumin on testicular extrinsic apoptosis pathway regulators (**a**) protein expressions in each group, (**b**) caspase 8, (**c**) cleaved-caspase 8 (**d**) caspase 3 (**e**) cleaved-caspase 3, (**f**) PARP, (**g**) cleaved-PARP in male mice. Data are expressed as means ± SD. n = 6–9 per group. Bars (a, b) with different letters indicate that values are significantly different. (One-way ANOVA, Duncan’s test, *p* < 0.05). AIN-93M: AIN-93M diet; LC: low carbohydrate diet; LC/AIN-93M: low carbohydrate diet changed to AIN-93M diet; LC/KD: low carbohydrate diet changed to ketogenic diet; LC/KDCu: low carbohydrate diet changed to ketogenic diet and curcumin.

**Figure 10 ijms-23-10009-f010:**
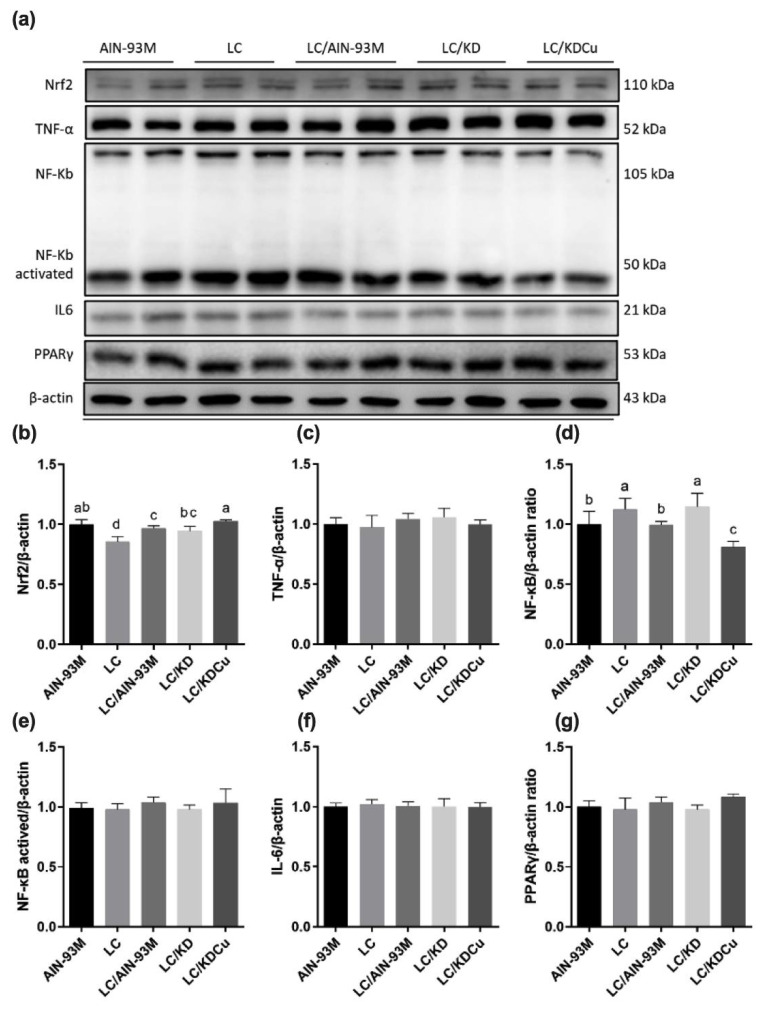
Effects of the low carbohydrate diet and curcumin on (**a**) protein expressions in each group, (**b**) Nrf2, (**c**) TNF-α, (**d**) NF-κB, (**e**) NF-κBactived, (**f**) IL-6, (**g**) PPARγin male mice. Data are expressed as means ± SD. n = 6–9 per group. Bars (a, b, c, d) with different letters indicate that values are significantly different. (One-way ANOVA, Duncan’s test, *p* < 0.05). AIN-93M: AIN-93M diet; LC: low carbohydrate diet; LC/AIN-93M: low carbohydrate diet changed to AIN-93M diet; LC/KD: low carbohydrate diet changed to ketogenic diet; LC/KDCu: low carbohydrate diet changed to ketogenic diet and curcumin.

**Table 1 ijms-23-10009-t001:** Feed ingredients and energy composition of each diet.

Ingredient (g/L)	AIN-93M	LC	KD	KDCu
Casein	28.00	28.00	28.00	28.00
Methionine	0.30	0.30	0.30	0.30
L-cystine	0.50	0.50	0.50	0.50
Dextrin	192.00	22.00	12.00	12.00
Sucrose	-	23.00	-	-
Soybean oil	10.00	75.00	90.00	90.00
Cellulose	10.00	10.00	10.00	10.00
Xanthan gum	3.00	3.00	3.00	3.00
Mineral	15.00	15.00	15.00	15.00
Vitamin	5.00	5.00	5.00	5.00
Choline	0.53	0.53	0.53	0.53
Curcumin	-	-	-	80 mg/kg
Energy (% of kcal)	AIN-93M	LC	KD	KDCu
Carbohydrate	79.2	18.6	4.9	4.9
Fat	9.3	69.6	83.2	83.2
Protein	11.9	11.9	11.8	11.8

## Data Availability

All data generated in the study are presented in the manuscript.

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
