# Peer review of "Curcumin Remedies Testicular Function and Spermatogenesis in Male Mice with Low-Carbohydrate-Diet-Induced Metabolic Dysfunction"

_ijms, 2022, doi:10.3390/ijms231710009_

Round 1

Reviewer 1 Report

In the present manuscript, Tsao et al examined the effect of curcumin on testicular function and spermatogenesis in male mice with low carbohydrate diet-induced metabolic dysfunction. Authors reported that a ketogenic diet supplemented with curcumin increased the testosterone level and improved semen quality by reducing oxidative stress. The author divided mice into two different groups normal diet (AIN-93M group) and a low carbohydrate diet (LC group), after 12 weeks LC grouped animals were further divided into four different groups depending on their diet, original diet (LC group), supplemented with AIN-93M diet (LC/AIN-93M group), a ketogenic diet (LC/KD group), and a ketogenic diet supplemented with curcumin (LC/KDCu group) for the final 6 weeks. Histological studies were performed to understand testicular histology and thickness of the GE in mice. Serum analysis was carried out for analysing total cholesterol triglyceride, aspartate aminotransferase, and alanine aminotransferase levels. Sperm quality was assessed by analysing sperm motility, sperm count, and morphological abnormality. Testicular hormone analysis was carried out using an ELISA test. Protein expression of pro-apoptotic and anti-apoptotic factors was carried out using western blot

·       In the present study, the author used a dosage of 80 mg/kg of curcumin. It’s a known fact that the clinical effectiveness of curcumin is weak due to its low bioavailability and poor absorption.  What’s the rationale behind using the current dosage? Will it be clinically relevant?

·       What was the route of curcumin administration? Is it through oral gavage? Authors should mention it in the materials and methods sections.

·       Interestingly authors mentioned in the discussion section that the previous studies showed curcumin improves sperm motility, but in the present study (figure 2) it looks like curcumin didn’t improve sperm motility. Authors should discuss it properly in the discussion section

·       Authors should label the figures according to figure legend and Check the paragraph and line spacing

Author Response

Reviewer 1

Comments and Suggestions for Authors

In the present manuscript, Tsao et al examined the effect of curcumin on testicular function and spermatogenesis in male mice with low carbohydrate diet-induced metabolic dysfunction. Authors reported that a ketogenic diet supplemented with curcumin increased the testosterone level and improved semen quality by reducing oxidative stress. The author divided mice into two different groups normal diet (AIN-93M group) and a low carbohydrate diet (LC group), after 12 weeks LC grouped animals were further divided into four different groups depending on their diet, original diet (LC group), supplemented with AIN-93M diet (LC/AIN-93M group), a ketogenic diet (LC/KD group), and a ketogenic diet supplemented with curcumin (LC/KDCu group) for the final 6 weeks. Histological studies were performed to understand testicular histology and thickness of the GE in mice. Serum analysis was carried out for analysing total cholesterol triglyceride, aspartate aminotransferase, and alanine aminotransferase levels. Sperm quality was assessed by analysing sperm motility, sperm count, and morphological abnormality. Testicular hormone analysis was carried out using an ELISA test. Protein expression of pro-apoptotic and anti-apoptotic factors was carried out using western blot.

  • In the present study, the author used a dosage of 80 mg/kg of curcumin. It’s a known fact that the clinical effectiveness of curcumin is weak due to its low bioavailability and poor absorption. What’s the rationale behind using the current dosage? Will it be clinically relevant?

Our response:

We thank the reviewer for the academic comment. The study shows that oral curcumin (80 mg/kg) can lower lipid accumulation in liver and adipose tissue and improved insulin sensitivity of male C57BL/6 mice with 60% high-fat diet through regulating SREBP pathway [1]. Prophylactic oral administration of curcumin (80 mg/kg) in Sprague Dawley rats with 60% high-fat diet feeding showed an anti-hyperglycemic, anti-lipolytic effects and anti-inflammatory by attenuating TNF-α levels [2]. In ICR mice with spermatogenic disorders induced by scrotal heat stress, administrating curcumin (80 mg/kg) by intragastric intubation also played the roles of antioxidative, anti-apoptotic and androgen synthesis effects [3]. In our study, the low carbohydrate diet and ketogenic diet contained 69.9% and 83.2% fat respectively to mimic the fashionable dietary patterns for weight control, we wanted to illustrate the effects of different dietary patterns and fat contains with or without curcumin on the testicular antioxidant capacity, apoptosis, inflammation and spermatogenesis, and our results showed that a low carbohydrate diet caused a lower sperm quality and a damaged testicular histology (figure 2 and 4). Administrating curcumin (80 mg/kg) could partial reverse this situation.

  • Interestingly authors mentioned in the discussion section that the previous studies showed curcumin improves sperm motility, but in the present study (figure 2) it looks like curcumin didn’t improve sperm motility. Authors should discuss it properly in the discussion section

Our response:

We thank the reviewer to point this out. It has been revised and the changes were highlighted in red color in the discussion.

Indeed the limiting factors of curcumin such as low solubility and oral bioavailability restrict its application as therapeutic agent [4,5]. To develop of oral delivery systems for curcumin to enhance the solubility and oral bioavailability such as nanoparticles [6] . The study shows that curcumin nanomicelle supplement (80 mg per day) to asthenoteratospermia [6] or oligozoospermia [7] patients could improve semen quality. In the present study, low carbohydrate diet impaired the sperm motility, morphology, MTBS and GE, and the ketogenic diet with curcumin improved the morphology, MTBS and GE but without motility. Bioavailability of curcumin could be improved by nano-curcumin formulations. Therefore, future investigation is suggested in this regard.

  • What was the route of curcumin administration? Is it through oral gavage? Authors should mention it in the materials and methods sections.

Our response:

We thank the reviewer to point this out. The curcumin powder was dissolved in soybean oil and added in the liquid diet, and we revised and highlighted in red color in the “2.2. Study design “section.

2.2. Study design

In the present study, after 1 week of acclimatization, the male C57BL/6 mice were randomly divided into two groups as follows: the AIN-93M group (n = 12) and the LC group (n = 48). After 12 weeks, the LC group was randomly divided into four groups: one was maintained on the original diet (LC, n = 12), the second changed to the AIN-93M diet (LC/AIN-93M, n = 12), the third changed to a ketogenic diet (LC/KD, n = 12), and the final group changed to a ketogenic diet with curcumin supplementation (LC/KDCu, n = 12) for 6 weeks. The curcumin powder was dissolved in the soybean oil and mixed in the liquid diet. The ingredients of each diet are shown in Table 1.

  • Authors should label the figures according to figure legend and Check the paragraph and line spacing
  • Our response:

We thank the reviewer to point this out. It has been revised and the changes were highlighted in red color.

Ref:

  1. Ding, L.; Li, J.; Song, B.; Xiao, X.; Zhang, B.; Qi, M.; Huang, W.; Yang, L.; Wang, Z. Curcumin rescues high fat diet-induced obesity and insulin sensitivity in mice through regulating SREBP pathway. Toxicol Appl Pharmacol 2016, 304, 99-109, doi:10.1016/j.taap.2016.05.011.
  2. El-Moselhy, M.A.; Taye, A.; Sharkawi, S.S.; El-Sisi, S.F.; Ahmed, A.F. The antihyperglycemic effect of curcumin in high fat diet fed rats. Role of TNF-alpha and free fatty acids. Food Chem Toxicol 2011, 49, 1129-1140, doi:10.1016/j.fct.2011.02.004.
  3. Lin, C.; Shin, D.G.; Park, S.G.; Chu, S.B.; Gwon, L.W.; Lee, J.G.; Yon, J.M.; Baek, I.J.; Nam, S.Y. Curcumin dose-dependently improves spermatogenic disorders induced by scrotal heat stress in mice. Food Funct 2015, 6, 3770-3777, doi:10.1039/c5fo00726g.
  4. Hewlings, S.J.; Kalman, D.S. Curcumin: A Review of Its Effects on Human Health. Foods 2017, 6, doi:10.3390/foods6100092.
  5. Liu, W.; Zhai, Y.; Heng, X.; Che, F.Y.; Chen, W.; Sun, D.; Zhai, G. Oral bioavailability of curcumin: problems and advancements. J Drug Target 2016, 24, 694-702, doi:10.3109/1061186X.2016.1157883.
  6. Rahimi, H.R.; Nedaeinia, R.; Sepehri Shamloo, A.; Nikdoust, S.; Kazemi Oskuee, R. Novel delivery system for natural products: Nano-curcumin formulations. Avicenna J Phytomed 2016, 6, 383-398.
  7. Alizadeh, F.; Javadi, M.; Karami, A.A.; Gholaminejad, F.; Kavianpour, M.; Haghighian, H.K. Curcumin nanomicelle improves semen parameters, oxidative stress, inflammatory biomarkers, and reproductive hormones in infertile men: A randomized clinical trial. Phytother Res 2018, 32, 514-521, doi:10.1002/ptr.5998.

Reviewer 2 Report

1. The authors use the terms sugar and glucose interchangeably (section 2.3 and figure 1). It is not correct. It would be appropriate not to use the term "sugar" when it comes to a specific type of sugar (in this case, glucose).

2. In the first quoted publication, the author's name was not mentioned.

3. References are cited chaotically. The authors do not maintain a uniform citation format.

Author Response

Reviewer 2

Comments and Suggestions for Authors

This is an interesting study focused on the effect of diet in male fertility. The research is well-conducted, and the results obtained of great relevance. Nevertheless, I have some comments that authors must address before the acceptance of the manuscript for publication:

  • The Abstract section needs some improvement to better understand the experimental design of the study (lines 13-16).

Our response:

Thank you for this advice. We have revised the abstract and made changes to the text throughout, which are marked in red.

Abstract: Increasing reports on the significance of dietary patterns in reproduction is arising from both animal and human studies, suggesting an interactive association between nutrition and male fertility. The aim of this study was to investigate the effects of curcumin supplementation on low carbohydrate diet-induced metabolic dysfunction, testicular antioxidant capacity, apoptosis, inflammation and spermatogenesis in male mice.  Male C57BL/6 mice were fed a normal diet (AIN-93M group, n=12) and a low carbohydrate diet (LC group, fed with low carbohydrate diet, n=48) for 12 weeks and mice randomly chosen from the LC group were later kept original diet (LC group, n=12), changed into AIN-93M feed (LC/AIN-93M group, n=12) or ketogenic diet (LC/KD group, n=12) or ketogenic diet treated with curcumin supplementation (LC/KDCu group, n=12) for the final 6 weeks. A poor sperm morphology and mean testicular biopsy score (MTBS) were observed in the LC and LC/KD groups, but they were eliminated by the normal diet or ketogenic diet with curcumin. The LC group exhibited a lower testicular testosterone level and a lower 17β-HSD activity and protein expression. It also enhanced apoptosis protein expressions in testis tissue, including Bax/BCl2, cleaved caspase 3, PARP and NF-κB. Meanwhile we found a statistically significant increase in lipid peroxidation and decreased superoxide dismutase (SOD), catalase (CAT) and glutathione peroxidase levels in the LC group. Our study indicated that replaced with a normal diet or ketogenic diet supplemented with curcumin attenuated the poor semen quality and reduced testosterone level by the LC diet via reducing oxidative stress.

  • Lines 104-105. Authors must provide a more detailed explanation of the procedure.

Our response:

We thank the reviewer to point this out. It has been revised and the changes were highlighted in red color in the “2.3. Histological analysis “section.

2.3. Histological analysis

Formalin-fixed testis tissues were prepared and analyzed at the Department of Pathology of Cardinal Tien Hospital (New Taipei City, Taiwan), being embedded, cut, and stained with Hematoxylin and Eosin (H&E) following standard protocols. The samples were observed and captured under a DM1000 microscope (Leica, Wetzlar, Germany). The thickness of the testicular germinal epithelium (GE) and the mean seminiferous tubule diameter (MSTD) were calculated across axes using Image J software (1.50, National Institutes of Health, Bethesda, MD, USA). The GE size was measured using the average length of a lumen and the length of a seminiferous tubule, and at least 10 sections were calculated in each mouse. Johnsen’s score was used to evaluate testicular spermatogenesis. The degree of testicular damages was tested using a 1–10 point’s scale ranging from the highest score of 10, standing fot complete spermatogenesis, to 1, indicating no seminiferous epithelium.

  • Line 124. Please, refer to “morphologically normal sperm” instead of “normal spermatozoa”.

Our response:

We thank the reviewer to point this out. It has been revised and the changes were highlighted in red color in the “2.5. Sperm quality analysis “section.

2.5. Sperm quality analysis

Sperm quality was assessed according to sperm motility, sperm count, and morphological abnormality. Sperm motility is expressed as the percentage of motile sperm evaluated microscopically under a magnification of 40× in 4 random microscopic fields, and at least 200 sperm cells were counted. The sperm count was measured using an automated cell counter (TC20, Bio-Rad, Hercules, CA, USA).

For sperm morphology, sperm slides were fixed with methanol (Honeywell, Morris Plains, NJ, USA) and stained with an eosin Y (E4009, Sigma-Aldrich, Saint Louis, MO, USA) and ethanol (Bioman, Taipei City, Taiwan) mixture. Then, the slides were rinsed with 75% ethanol (Bioman) and dried. Finally, the percentage of morphologically normal sperm within at least 100 sperm cells was calculated.

  • Subsection 2.4. Please, the measure units of each serum parameter.

Our response:

We thank the reviewer to point this out. It has been revised and the changes were highlighted in red color in the “2.4. Serum analysis “section.

2.4. Serum analysis

After anaesthetization, blood from heart was sit for 20-30 min at room temperature before being centrifuged for 15 min at 3000 rpm and separated the resulting supernatant. Total cholesterol (TC, mg/dl), triglyceride (TG, mg/dl), high-density lipoprotein (HDL, mg/dl), aspartate aminotransferase (AST, U/l), and alanine aminotransferase (ALT, U/l) levels were measured using a chemistry analyzer (Modular P800, Roche, Basel, Switzerland). Serum ketone (mmol/l) and glucose (mg/dl) levels were measured using a GLUCOSURE POC system (ApexBio AB-302G, Hsinchu, Taipei). The serum insulin level (pmol/l) was determined by Enzyme-Linked Immunosorbent Assay (ELISA) using a commercial ELISA kit according to the manufacturer’s instructions (Mercodia Mouse Insulin ELISA, 10-1247-01, Uppsala, Sweden).

  • Figures 1 to 10. Please, add the appropriate letter to each plot and/or image to improve the readability of each figure legend.

Our response:

 Thank you for pointing out this omission. All the figure legends have been revised accordingly, and the changes are shown in red. Data were analyzed using one-way ANOVA and Duncan’s post hoc test to ascertain which mean values differed between treatment groups. Bars with different letters indicate that values are significantly different (p < 0.05)

Taking the example of Figure 8, there were no significant differences among the AIM-93M, LC and LC/KD groups (bars marked with “a” and “ab”). The LC/KD/Cu, LC/AIN-93M, AIN-93M and LC/KD groups (bars marked with “b” or “ab”) also did not differ significantly. The LC/AIN-93M and LC/KD/Cu groups values (bars with “b” ) were significantly reduced in comparison with the LC group.

  • Line 217 “thickness of GE”. Authors must provide the results of their measures. Besides, they must add a detailed explanation in the Material and Method section describing how these measures were performed.

Our response:

We thank the reviewer to point this out. The measurement of GE has been revised and the changes were highlighted in red color in the “2.3. Histological analysis “section.

2.3. Histological analysis

Formalin-fixed testis tissues were prepared and analyzed at the Department of Pathology of Cardinal Tien Hospital (New Taipei City, Taiwan), being embedded, cut, and stained with Hematoxylin and Eosin (H&E) following standard protocols. The samples were observed and captured under a DM1000 microscope (Leica, Wetzlar, Germany). The thickness of the testicular germinal epithelium (GE) and the mean seminiferous tubule diameter (MSTD) were calculated across axes using Image J software (1.50, National Institutes of Health, Bethesda, MD, USA). The GE size was measured using the average length of a lumen and the length of a seminiferous tubule, and at least 10 sections were calculated in each mouse. Johnsen’s score was used to evaluate testicular spermatogenesis. The degree of testicular damages was tested using a 1–10 point’s scale ranging from the highest score of 10, standing for complete spermatogenesis, to 1, indicating no seminiferous epithelium.

The results of thickness of GE were 59.41 ± 4.01 (AIN-93M group), 42.36 ± 4.08 (LC group), 58.22 ± 4.29 (LC/AIN-93M), 48.21 ± 1.90 (LC/KD) and 57.61 ± 3.33. (LC/KDCu) respectively. These changes were also highlighted in red color in the “3.2. Curcumin ameliorates the decrease in testicular spermatogenesis and sperm quality caused by a low carbohydrate diet “section.

3.2. Curcumin ameliorates the decrease in testicular spermatogenesis and sperm quality caused by a low carbohydrate diet

The results showed that there were no significant differences in the weights of the testis, epididymis or vas deferens among the five groups. In terms of sperm motility, the AIN-93M group was significantly better than the other four groups, and as regards sperm count, there were no significant differences among the five groups. In the case of sperm morphology, the percentage of morphologically normal sperm in the LC group was significantly lower than in the other four groups (Figure 2).

Figure 3 illustrates the correlation between ketone bodies and sperm quality. Correlation results included: sperm motility (p = 0.0043, r = 0.1490), sperm count (p = 0.2037, r = 0.0292), and sperm morphology (p = 0.7551, r = 0.0055).

Figure 4 shows that there was no significant difference in the MSTD among the five groups. The MTBS of the LC and the LC changed to the ketogenic diet (LC/KD) groups was significantly lower than that of the AIN-93M and LC/KDCu groups, and the thickness of the GE in the LC and LC/KD groups was significantly lower than in the other three groups. The results of thickness of GE were 59.41 ± 4.01 (AIN-93M group), 42.36 ± 4.08 (LC group), 58.22 ± 4.29 (LC/AIN-93M), 48.21 ± 1.90 (LC/KD) and 57.61 ± 3.33. (LC/KDCu) respectively. To sum up, the LC/AIN-93M and the LC/KDCu groups had better sperm production and improved spermatogenesis.

Reviewer 3 Report

This is an interesting study focused on the effect of diet in male fertility. The research is well-conducted, and the results obtained of great relevance. Nevertheless, I have some comments that authors must address before the acceptance of the manuscript for publication:

1)     The Abstract section needs some improvement to better understand the experimental design of the study (lines 13-16).

2)     Lines 104-105. Authors must provide a more detailed explanation of the procedure.

3)     Line 124. Please, refer to “morphologically normal sperm” instead of “normal spermatozoa”.

4)     Subsection 2.4. Please, the measure units of each serum parameter.

5)     Figures 1 to 10. Please, add the appropriate letter to each plot and/or image to improve the readability of each figure legend.

6)     Line 217 “thickness of GE”. Authors must provide the results of their measures. Besides, they must add a detailed explanation in the Material and Method section describing how these measures were performed.

Author Response

Reviewer 3

  • The authors use the terms sugar and glucose interchangeably (section 2.4 and figure 1). It is not correct. It would be appropriate not to use the term "sugar" when it comes to a specific type of sugar (in this case, glucose).
  • Our response:
  • We thank the reviewer to point this out. It has been revised and the changes were highlighted in red color of figure and in the “4. Serum analysis “section.

2.4. Serum analysis

After anaesthetization, blood from heart was sit for 20-30 min at room temperature before being centrifuged for 15 min at 3000 rpm and separated the resulting supernatant. Total cholesterol (TC, mg/dl), triglyceride (TG, mg/dl), high-density lipoprotein (HDL, mg/dl), aspartate aminotransferase (AST, U/l), and alanine aminotransferase (ALT, U/l) levels were measured using a chemistry analyzer (Modular P800, Roche, Basel, Switzerland). Serum ketone (mmol/l) and glucose (mg/dl) levels were measured using a GLUCOSURE POC system (ApexBio AB-302G, Hsinchu, Taipei). The serum insulin level (pmol/l) was determined by Enzyme-Linked Immunosorbent Assay (ELISA) using a commercial ELISA kit according to the manufacturer’s instructions (Mercodia Mouse Insulin ELISA, 10-1247-01, Uppsala, Sweden).

Figure 1. Effects of the low carbohydrate diet and curcumin on (a) body weight, (b) food intake, (c) serum total cholesterol (TC), (d)serum triglycerides (TG), (e) serum high density lipoprotein (HDL), (f) serum aspartate aminotransferase (AST), (g) serum alanine aminotransferase (ALT), (h) serum glucose in male mice: Data are expressed as means ± SD. ((a-b) n=11-12; (c-h) n=7-12 per group). a, b Bars with different letters indicate that values are significantly different. (One-way ANOVA, Duncan's test, p < 0.05). AIN-93M: AIN-93M diet; LC: low carbohydrate diet; LC/AIN-93M: low carbohydrate diet changed to AIN-93M diet; LC/KD: low carbohydrate diet changed to ketogenic diet; LC/KDCu: low carbohydrate diet changed to ketogenic diet and curcumin.

  • In the first quoted publication, the author's name was not mentioned.

Our response:

Thank you for pointing out this omission. All the reference have been checked revised.

  • References are cited chaotically. The authors do not maintain a uniform citation format.

Our response:

Thank you for pointing it out. All the reference have been revised and changed the citation style to the ACS as recommended.

Reviewer 4 Report

The manuscript ‘Curcumin Remedies Testicular Function and Spermatogenesis In Male Mice With Low Carbohydrate Diet-Induced Metabolic Dysfunction’ is well organized and the topic indeed interesting and attractive, entirely falls into the scopes of the IJMS if the author considers the opinion could be addressed to improve the paper:

1. In table 1 diet composition need to be expressed with the unit.

2. Treatment dose is not clear; the author needs to be clear about dosages amount and times.

3. Section needs to be rearranged, please put material methods in section 2 instead of results or abbreviated all in section 2.

4. There is grammatical mistake throughout the manuscript.

Author Response

Comments and Suggestions for Authors

The manuscript ‘Curcumin Remedies Testicular Function and Spermatogenesis In Male Mice With Low Carbohydrate Diet-Induced Metabolic Dysfunction’ is well organized and the topic indeed interesting and attractive, entirely falls into the scopes of the IJMS if the author considers the opinion could be addressed to improve the paper:

  1. In table 1 diet composition need to be expressed with the unit.

Our response:

We thank the reviewer for the opinion. The unit of diet composition is g/L that can be found beside the ingredient of the first row in table 1.

  1. Treatment dose is not clear; the author needs to be clear about dosages amount and times.

Our response:

We thank the reviewer for the comment. The treatment was dissolved in soybean oil and the dose was 80 mg/kgBW/day, and the related description was added in the 4.2 study design of materials and methods.

  1. Section needs to be rearranged, please put material methods in section 2 instead of results or abbreviated all in section 2.

Our response:

We thank the reviewer for the query. We have changed the order of the manuscript based on the journal template according to the editor’s suggestions during last revising process. Herein we put the “Results” in section 2, the “Discussion” in section 3 and the “Materials and Methods” in section 4 finally.

  1. There is grammatical mistake throughout the manuscript.

Our response:

We thank the reviewer for the suggestion. The manuscript was revised by MDPI English language editing finally.

Round 2

Author Response

Dear the reviewer:

We have revised the manuscript according to the referee' comment. Thanks for your help and attention !!

Chih-Wei Tsao/Professor Liu

Reviewer 3 Report

Authors have introduced all the modifications requested by the reviewers and so the overall quality of the manuscript has been improved.

Author Response

(The authors gave the same response as above.)
